# Tuberculosis in an Aging World

**DOI:** 10.3390/pathogens11101101

**Published:** 2022-09-26

**Authors:** Angélica M. Olmo-Fontánez, Joanne Turner

**Affiliations:** 1Host Pathogen Interactions and Population Health Programs, Texas Biomedical Research Institute, San Antonio, TX 78227, USA; 2Integrated Biomedical Sciences Program, University of Texas Health Science Center, San Antonio, TX 78229, USA

**Keywords:** tuberculosis, aging, *M. tb*, elderly, inflammaging, comorbidities

## Abstract

Tuberculosis (TB) is one of the leading causes of death due to its being an infectious disease, caused by the airborne pathogen *Mycobacterium tuberculosis* (*M.tb*). Approximately one-fourth of the world’s population is infected with latent *M.tb*, and TB is considered a global threat killing over 4000 people every day. The risk of TB susceptibility and mortality is significantly increased in individuals aged 65 and older, confirming that the elderly represent one of the largest reservoirs for *M.tb* infection. The elderly population faces many challenges that increase their risk of developing respiratory diseases, including TB. The challenges the elderly face in this regard include the following: decreased lung function, immuno-senescence, inflammaging, adverse drug effects, low tolerance to anti-TB drugs, lack of suitable diagnoses/interventions, and age-associated comorbidities. In order to find new therapeutic strategies to maintain lung homeostasis and resistance to respiratory infections as we age, it is necessary to understand the molecular and cellular mechanisms behind natural lung aging. This review focuses primarily on why the elderly are more susceptible to TB disease and death, with a focus on pulmonary function and comorbidities.

## 1. Tuberculosis in the Elderly

Tuberculosis (TB) has scourged humankind through the centuries. It is considered a global threat claiming more than 4000 lives every day, resulting in approximately 1.3 million deaths in 2020 [1,2]. This has, in part, been aggravated by the COVID-19 pandemic, which restricted access to TB diagnosis and treatment [3,4,5]. TB is a chronic inflammatory disease caused by the bacillus *Mycobacterium tuberculosis* (*M.tb*), which is easily spread from one person to another by airborne droplet nuclei. These aerosolized particles transit past the nasopharyngeal region to be delivered into the distal airways, e.g., the alveolus [6]. Although *M.tb* typically affects the lungs (pulmonary TB), other organs can also be vulnerable (extrapulmonary TB). Increased risk of TB disease and mortality are associated with populations that have compromised immunity, such as HIV-infected individuals, infants, and the elderly [7,8,9]. The main focus of this review is to discuss why the elderly are more susceptible to TB disease and death.

The elderly population (discussed here as ≥65 years old) is expected to double to 2 billion by 2050 [10,11] and increase to almost 95 million by 2060 in the United States alone [12]. The elderly population is at remarkable risk of developing respiratory diseases, including TB [7,13]. The global number of TB cases and incidence in the U.S. is higher in populations over 50 years of age with an overall male:female ratio of 2:1 (2.7 per 100,000 persons: 1.7 per 100,000 persons, respectively) [14]. The elderly are considered a large reservoir for *M.tb* infection, due to their increased susceptibility to new infections and reactivation of latent *M.tb* infection [7,15]. Additionally, medical care for the elderly can be challenging, due to their predisposition for chronic comorbidities and adverse drug reactions, leading to poorly managed anti-tuberculosis treatment combined with an increased mortality rate [16,17,18,19], predominantly because of waning immunity.

## 2. Lung Function Changes in the Elderly

Natural aging of the lung is associated with continuous changes at the molecular and physiological levels, leading to a persistent low-grade pro-inflammatory and oxidative state, decreased lung function and impaired immune responses, which increases susceptibility to lung disease and respiratory infections [13,20,21]. Impaired lung physiology with increasing age is due to significant changes in cellular and structural composition across the respiratory epithelium. As we age, lung function progressively declines, including reduced respiratory muscle strength, force expiratory capacity, and vital capacity [22,23,24,25]. Importantly, lung tissue from elderly individuals has significantly decreased tissue-repair capacity compared to young individuals [26]. Therefore, age-associated changes in the alveolar space, such as decreased alveolar septae and elasticity and changes in the extracellular matrix, impair the ability of the lung environment to respond to chemical, mechanical, and biological stressors appropriately [11,27]. Moreover, aging leads to detrimental alterations in cellular function and cell-to-cell interactions of pulmonary resident cells and peripheral immune cells, leading to an imbalance and decline of appropriate innate and adaptive immune responses, associated with immuno-senescence [21,28,29]. Therefore, aging negatively interferes with critical physiological and immunological mechanisms to maintain the coordinated functions of multiple cell types to sustain cellular homeostasis. Additionally, hallmarks of cellular aging, such as increased mitochondrial dysfunction, oxidative stress, and cellular senescence, as well as decreased proteo-stasis and telomere length, contribute to the dysregulated regenerative capacity of lung tissue, inflammation, and immunosurveillance, leading to increased susceptibility to infection and development of chronic respiratory diseases [11,21]. Understanding the molecular and cellular mechanisms behind natural lung aging remains an important goal in the search for novel therapeutic approaches to preserve lung homeostasis as we age.

## 3. Inflammaging and/or Immuno-Senescence?

Cell-mediated immunity is the primary mechanism of immune protection against *M.tb* infection. However, changes in the lung environment as we age can modify the cellular function of both pulmonary innate and adaptive cells [21,27]. For example, changes occur in macrophages (decreased phagocytosis and increased superoxide production), neutrophils (decreased phagocytosis and dysregulated chemotaxis), dendritic cells (decreasing antigen presentation), and T lymphocytes (reduced naïve T cell number, impaired T cell helper responses and cytotoxic responses), amongst others [21,27,28] (Figure 1). It is necessary to better understand how increasing age influences the pulmonary environment at both molecular and cellular levels to appreciate how this can impact susceptibility to develop TB disease or mortality. One hallmark of the aging lung is a heightened basal level of pro-inflammatory mediators (termed inflammaging) [30]. Inflammaging was originally characterized as a persistent low-grade pro-inflammatory state in the circulation that progressively increases with age [31], interfering with regulation of anti-inflammatory responses, driving immuno-senescence, and increasing the risk of chronic diseases as an individual ages [30,32]. The molecular composition of the alveolar lining fluid (ALF) components (e.g., innate soluble protein/cytokines and lipids) in the lung are also significantly altered in aged mice and elderly humans, resulting in a relatively oxidized environment [27,33,34]. Such studies confirmed that inflammation is not limited to the periphery, but can also be found within the lung and, likely, within other tissues too. Related to *M.tb* infection control, the ALF of elderly humans can differentially alter the *M.tb* cell envelope surface upon contact [35,36,37], and alter the expression of genes involved in cell envelope remodeling, metabolism, and virulence [38]. Significantly, *M.tb* exposure to ALF from the elderly can increase bacterial virulence and pathogenicity in vivo in mice [39]. Mechanistically, age-associated functional alterations of ALF components in humans (e.g., decreased surfactant protein D [SP-D] binding) [33,39] are associated with decreased host capacity to control *M.tb* infection in vitro [39,40], and to replenish with functional SP-D so as to reestablish the capacity of human macrophages to control *M.tb* infection [39]. These findings confirm how the aging lung environment can alter soluble innate lung components and impair protective functions of the lung. 

Additional studies support these findings at the cellular level, where, although alveolar macrophages from old mice have elevated IFN-γ-induced activation markers, they show reduced control of *M.tb* infection [41]. Similarly, although *M.tb* infected macrophages from elderly individuals have a higher basal activation and ROS production, the cells are more permissive to intracellular growth [13]. Furthermore, characterization of alveolar macrophage subpopulations in old mice with unique inflammatory or regulatory signatures demonstrate that the aged inflammatory macrophages are more permissive to *M.tb* growth and survival [42]. Neutrophils are rapidly recruited to the mycobacterial infection site, and their phagocytic function includes a broad range of antimicrobial properties (ROS, cytotoxic proteases, and neutrophil extracellular traps [NETs]) to potentially clear infection [43]. Although aging impacts neutrophil function by decreasing NET formation and phagocytosis, as well as causing dysregulated accumulation and chemotaxis [21,44], the overall data about the microbicidal activity of neutrophils are often conflicting [45], and more studies are needed to understand how aged neutrophils control *M.tb* infection. Aged mice showed resistance to *M.tb* infection associated with an early and transient inflammatory environment [46,47,48] and, thus, neutrophils and CD11b^+^ cells might contribute to early protection. Partial depletion of neutrophils in lipopolysaccharide-treated mice (a model for inflammation in old age) demonstrated decreased *M.tb* association and killing [49]. The prolonged recruitment and activation (degranulation) of neutrophils in older individuals in response to infection likely promotes chronic inflammation in the lung epithelium (tissue damage) and contributes to the susceptibility of the elderly to develop, or succumb to, TB. 

Altered T cell-mediated immunity is associated with advanced aging, which results in decreased capacity to control *M.tb* infection [50]. Importantly, the emergence of protective CD4^+^ T cells was shown to be delayed in old mice [51], suggesting a delayed accumulation and migration of CD4^+^ T cells of old mice into *M.tb* infection sites. Adaptive CD4^+^ and CD8^+^ T-cell compartments become compromised with age, but, despite this, old mice were able to express a transient early resistance to *M.tb* that was mediated by innate CD8^+^ T cells that could respond to IL-12 in an antigen-MHC-I independent manner and secrete IFN-γ [46,48,52]. This early IFN-γ response was sufficient to limit the initial growth of *M.tb*. However, old mice are unable to generate sufficient IL-2-secreting antigen-specific T cells due to low proliferative capacity within the lungs [53,54,55], limiting the ability of old mice to combat *M.tb* infection. Overall, chronic inflammation during aging hinders T-cell responses and interferes with vaccine efficacy [56,57,58]. For example, applying a delayed-type hypersensitivity model of *Mycobacterium bovis* BCG vaccination and tuberculin skin test in old baboons showed impaired immune responses to antigenic challenges that varied between tissue sites and the periphery, limiting appropriate immune memory responses [59]. A follow-up study demonstrated a decrease or delay in T cell recall responses to the pulmonary challenge site in aged BCG vaccinated rhesus macaques [60]. In contrast, BCG (strain Tokyo 172) revaccination of guinea pigs (young, middle-aged, and old groups) showed reduced bacterial growth in different organs, demonstrating the importance of a BCG booster to confer protection, regardless of age [61]. Although BCG-inoculated aged mice showed protection against *M.tb* infection comparable to young mice [62], the efficacy was eventually lost over time [63].

## 4. How Comorbidities of Aging Impact TB in the Elderly

Although many studies have characterized how aging-associated changes in innate and adaptive immunity can be key drivers for the increased susceptibility of the elderly to respiratory infections, it remains unclear how age-associated comorbidities may contribute to this elevated risk. Specific for TB, numerous age-related comorbidities, such as diabetes, obesity, malnutrition, chronic respiratory diseases, cancer, and other underlying medical conditions, can result in an increased risk for developing active TB disease [7,8]. As a result, absolute mortality increases as the number of comorbidities and age increases [64].

A meta-analysis of the association between diabetes and active TB in several observational studies revealed that Diabetes mellitus (DM) increases the risk for active TB by approximately three-fold in all age groups [65]. The prevalence of DM increases with age due to baseline glucose tolerance being lower in the elderly (even without DM) [66], leading to an elevated risk of TB compared to the younger population. The authors even concluded that in countries such as India and China, DM may already be responsible for more than 10% of active TB cases [65]. Patients with DM and TB have an increased risk of TB treatment failure and death compared to TB patients without DM [67]. There is a premise that diabetes directly compromises innate and adaptive immune responses [68]. DM causes impaired neutrophil function (chemotaxis, adherence and phagocytosis), and suppression of cytokine production, as well as reducing non-specific IFN-γ production, which is critical for the initial stages of *M.tb* recognition [68,69,70]. In contrast, a new cross-sectional study revealed unique findings on the association between DM, TB, BCG vaccination, and aging. For instance, DM was not a risk factor in the Hispanic elderly (>60 years) community, despite its high prevalence [71]. An additional interesting finding was that BCG vaccination conferred a protective effect for TB in the elderly [71]. In summary, DM is considered a significant contributing risk factor for TB, mainly in younger individuals; however, as we get older that risk is reduced. 

An additional metabolic disorder linked to DM is obesity, which negatively contributes to the inflammation and impairment of cell-mediated immunity. Increased levels of white adipose tissue secrete significant pro-inflammatory cytokines (TNFα, CRP, interleukins), leading to chronic, low-grade inflammation, that is aggravated by the activation of complement components (C3, C3a and C3aR, C3adesArg [ASP] and Factor D) [72]. Although complement activation is considered a supportive component of our first line of defense against pathogens, its detrimental (bystander) activation might lead to increased tissue damage and susceptibility to TB. *M.tb* infected obese mice on a high-fat diet (HFD) demonstrate increased pulmonary inflammation, IFN-γ-mediated immunopathology, and susceptibility to *M.tb* infection [73]. Data that link aging, obesity, and TB are so far conflicting. A study in Panama showed that elderly and obese “household close contacts” were at higher risk of acquiring latent TB [74]. In contrast, an elderly and obese population in Hong Kong demonstrated a lower risk of active TB disease [75]. Either way, we can conclude that major health risk factors (e.g., obesity, DM) can influence optimum immune responses. 

The intake of micronutrients is critical for major metabolic pathways and immune cell functions [76]. Therefore, malnutrition is one of the most common causes of immunodeficiency worldwide and is considered an important risk factor for TB [77,78]. For example, zinc deficiency was associated with lymphoid atrophy, as well as impaired T-lymphocyte functions, macrophage migration, and cytokine production in BCG-vaccinated guinea pigs [79,80]. An additional study of BCG-vaccinated guinea pigs with Vitamin D3 (calcitriol) deficiency showed no effect on the course of TB disease in a non-vaccinated group, but considerable loss of T-cell capabilities in a BCG-vaccinated group [81]. In addition, *M.tb* infected macrophages from malnourished animals co-cultured with lymphocytes produced less TNF [82]. Elderly individuals, mainly those who are socially isolated, are at high risk of malnutrition, leading to malabsorption of essential nutrients necessary to stimulate adequate immune responses [83]. For instance, malnourished aged people demonstrate decreased CD4^+^ T cell subsets, monocytes and PMN density and, consequently, decreased release of IL-1 and IL-2 cytokines, among others [84,85,86]. This is partially due to the significantly low levels of albumin in malnourished elderly [84,87]. Furthermore, cross-sectional studies showed that TB patients suffered from nutrient deficiencies in vitamins (A, B_6_, D, and E), thiamin, and folate, among others, suggesting that TB can directly contribute to malnutrition [88,89]. There is a strong connection between malnutrition and immune impairment mainly mediated by T cells and T cells are already compromised in the elderly population, highlighting malnutrition as a factor that may aggravate the TB disease outcome in this population.

In addition to metabolic disorders, several chronic lung and kidney diseases may also represent risk factors for TB, including chronic obstructive pulmonary disease (COPD), idiopathic pulmonary fibrosis (IPF), lung cancer, and chronic kidney disease (CKD). The pathological features of these conditions are mainly caused by the detrimental role of aging that drives uncontrolled inflammation and abnormal tissue repair capabilities, consequently contributing in some cases to emphysema and pneumonia [90,91]. COPD is a lung disease characterized by chronic airflow obstruction in response to environmental factors (mainly cigarette smoke or wood burning fires) resulting in an irreversible inflammatory state and lung tissue remodeling [91]. When compared to the general population, COPD patients have a higher risk of developing active TB disease [92,93]. The authors anticipate that the global burden of COPD will raise the incidence of active TB [92]. Another detrimental effect of inflammaging is excessive activation of alveolar epithelial cells, fibroblasts and myofibroblasts, causing aberrant production of extracellular matrix, leading to IPF [94]. Patients with IPF have a higher TB incidence (five times greater) than the general population [95]. Although there are plenty of studies that attribute TB as a risk factor for lung cancer [96,97], we should investigate in depth how lung cancer can influence TB susceptibility in the elderly. It seems that cancer patients are more susceptible to TB than control groups [98,99]. A retrospective cohort study of elderly cancer patients demonstrated an increased risk of reactivating latent TB infections, but it was not exclusively related to lung cancer. It was also associated with other cancer types, including colon, oral and hematologic [99]. Lastly, CKD patients, particularly those undergoing dialysis, have a significantly higher risk of pulmonary TB compared to the general population, mainly due to their immunosuppressed status [100,101,102,103]. Although more studies are needed to understand how aging may exacerbate the impact of COPD, IPF, cancer or CKD in the progression of TB disease, undoubtedly, all age-associated comorbidities have one thing in common, and these are worsening immune system responses and inflammation, leading to an increased risk of TB disease.

## 5. Sociocultural Determinants Surpass Any Age Barrier

Socioeconomic deprivation (the disadvantage of an individual due to lack of economic and social resources), such as poor living conditions (homelessness), overcrowding, illicit drug and alcohol use, unemployment (or having a low income), imprisonment, and lack of education might contribute to TB susceptibility or risk of poor treatment outcomes [104,105]. There is increased TB prevalence when people are living in crowded conditions with poor ventilation, especially elderly individuals living in institutional care (nursing homes) [106]. Undoubtedly, people with poor socioeconomic status have an increased risk for TB, and it is accepted that social intervention, such as providing adequate nutrition and health care, including preventative therapy for latent TB patients, and avoidance of overcrowding, can prevent disease dissemination [104,105,107]. Thus, if we know what could be a feasible solution to stop the spread, why does TB remain a problem? It is likely due to challenges in TB surveillance programs (treatment supply and monitoring), or insufficient collaboration between the health sector and the community. Either way, we must overcome these challenges in order to control the current global TB epidemic.

## 6. Conclusions

Aging by itself is a very significant risk factor for developing TB. The continuous accumulation of basal inflammation and an oxidative state as we age (inflammaging) exacerbates the homeostatic balance of stress responses, impairing the intrinsic mechanisms that aid cell regeneration, repair, and immunosurveillance [11,21,27]. These outcomes consequently increase susceptibility to acute and chronic diseases and result in increased morbidity and mortality in the elderly [11]. Aging is a highly complex and dynamic process; thus, understanding fully how to prevent loss of, or restore, inflammatory reactivity and detrimental effects remains elusive. Even more complicated, is the addition of all the age-related comorbidities including DM, malnutrition, obesity, and HIV, that are currently driving the global TB epidemic (Figure 2) [1]. Alternative perspectives about how aging and TB are interconnected must also be evaluated. A recent study showed that TB infection increased epigenetic perturbations (DNA methylation) that correlated with inflammation and oxidative stress leading to premature cellular aging and increasing the risk of death in TB patients [108]. It is critical to determine strategies to reduce age-related inflammation to improve resistance to TB disease and quality of life in the elderly. 

BCG vaccination confers protection and controls inflammation by lowering plasma levels of types 1, 2, and 17 pro-inflammatory cytokines and type 1 interferon in elderly persons residing in hot spots for SARS-CoV-2 infection [109]. Additionally, BCG improved myeloid and T cell responsiveness in elderly individuals in [109]. A striking finding could be associated with reversal of inflammaging and immuno-senescence in the elderly [110] to potentially provide appropriate protection against other respiratory diseases, such as TB. Antioxidant and micronutrient supplementation (vitamin C, vitamin E, and nicotinamide adenine dinucleotide [NADH]) significantly improve T cell proliferation and responses, as well as control cellular oxidative stress in aging [111,112,113]. Thus, oral supplementation of these vitamins may represent a feasible and promising therapeutic strategy to manage cellular oxidative stress and age-related disorders. Interestingly, vitamin consumption (A, C, D, and E), in addition to providing antimycobacterial properties, may reduce TB risk in smokers, especially Vitamin C, by lowering levels of oxidative stress [114,115].

Lifestyle interventions, including diet (Caloric restriction [CR]), exercise training or avoiding frequent use of alcohol, drugs, and cigarette smoking, may provide lasting alternatives to control inflammation and immunological protection in the elderly. The compelling topic of moderate but not acute CR (by reducing plasma levels of glucose and insulin) associated with extending lifespan in some animal models is still feasible [116,117,118], although emerging evidence debates whether CR-related increased lifespan is universal in all mammalian species or not [119]. Overall, lowering glucose plasma levels by restricting food intake reduced metabolic rate and, consequently, the production of reactive oxygen species that may cause oxidative damage [118]. In addition, increased glucose and insulin levels (hyperglycemia and hyperinsulinemia) can cause molecular damage by glycation and glycoxidation, which is considered an age-related effect [120]. In addition to type of diet, physical fitness may contribute to the metabolic health of the elderly. Older women taking part in regular exercise training showed decreased inflammation and oxidative stress markers in adipose tissue [121,122]. Conversely, the absence of exercise training in women demonstrated a more pro-inflammatory environment, showing higher levels of TNFα and IL-8 cytokines and TH 1 cell recruitment [121]. Certainly, CR and exercise training may combat low-grade inflammation found in aging.

It is critical to consider how pharmacological interventions could exacerbate TB disease in the elderly. Adverse effects of anti-TB drugs are significant in elderly patients, due to the high incidence of underlying illnesses. Overall, elderly TB patients tolerate anti-TB medicines less well compared to young individuals, which reduces the effectiveness of anti-TB therapy and can cause neurotoxicity, skin lesions, arthralgia, hepatitis, and gastrointestinal discomfort, among others [16,123]. The main challenge remains in trying to avoid the sensitivity to drug reactions in a population with comorbidities that already require the administration of multiple drugs or treatments (polypharmacy), leading to a lowered efficiency of renal and hepatic drug clearance [15]. 

To conclude, molecular and physiological changes accumulate in the lungs as we age, leading to a persistent low-grade pro-inflammatory and oxidative state. Consequently, these detrimental changes, led primarily by inflammaging and immuno-senescence, interfere with the homeostatic balance of stress responses, leading to increased oxidative stress, mitochondrial dysfunction, cell injury, decreased lung function, impaired immunosurveillance, and increased susceptibility to chronic diseases and respiratory diseases, such as TB. Waning immunity, underlying conditions (co-morbidities), and susceptibility to adverse drug effects, among other factors, make the elderly population at higher risk for TB diseases and mortality (Figure 2). Diagnosis and positive TB treatment outcomes are challenging in elderly patients. For example, TB lung lesions are frequently misdiagnosed as pneumonia [123], or the elderly present fewer classical TB symptoms (e.g., cough, fever, night sweats and weight loss) [124]. Unfortunately, this causes a significant delay in the clinical TB diagnosis for the elderly [124]. Although there is no diagnostic gold standard test to detect latent-TB infection in the elderly, IGRA tests have been used for the detection of IFN-γ production (in response to specific peptides from *M.tb*) over the tuberculin skin test (TST), due to its increased sensitivity [125]. Phenotypic and genotypic testing have been implemented for the rapid diagnosis of *M.tb* infection, especially in low- and middle-income countries with high burden disease rates [126], as these tests are urgently needed, especially after the emergence of multi-drug resistant (MDR) strains. Due to the challenges of diagnosis and treatment of TB in the elderly we should focus our attention on (1) determining mechanisms to reduce inflammation levels at an appropriate time to avoid TB progression, and (2) determining methods to prevent adverse drug effects to ensure successful TB treatment outcomes in this fragile and compromised population. 

## Figures and Tables

**Figure 1 pathogens-11-01101-f001:**
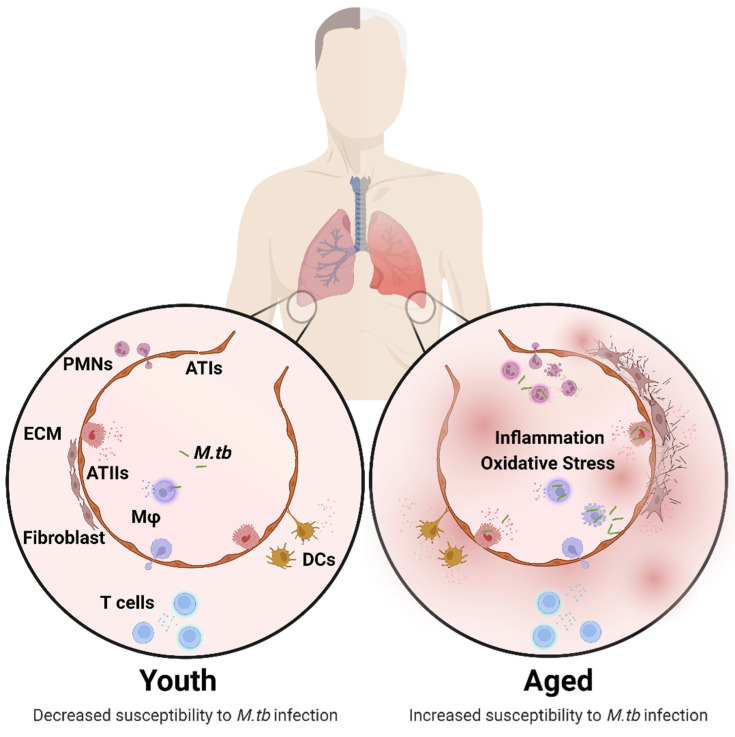
Structural and cellular changes in the aging lung. Age-associated changes in the composition and function of pulmonary cells is characterized by increased inflammation and oxidative stress driving impaired lung homeostasis, leading to increased susceptibility to *M.tb* infection. Structural and Cellular Changes: *Increased* alveolar space, inflammation, oxidative stress (ATs, Mφ), superoxide production (Mφ), apoptosis (ATs), fibrogenic responses, antigen presentation (DCs), CD4:CD8 ratio and senescence (ATs). *Decreased* mucociliary clearance, mucous production, alveolar septae, elasticity, phagocytosis (Mφ, PMNs, DCs), migration and proliferation (DCs, T cells), and naïve T cell number. *Changes* in ECM, surfactant composition (ATs), cytokine production, and chemotaxis (PMNs). A complete description of the cellular composition and functional changes in the aging lung is reviewed in detail elsewhere [11,21,27]. Abbreviations: ECM (extracellular membrane), ATs (alveolar epithelial cells), Mφ (macrophages), PMNs (neutrophils), DCs (dendritic cells), *M.tb* (*Mycobacterium tuberculosis*). This illustration was created with BioRender (https://biorender.com/), accessed on 18 September 2022.

**Figure 2 pathogens-11-01101-f002:**
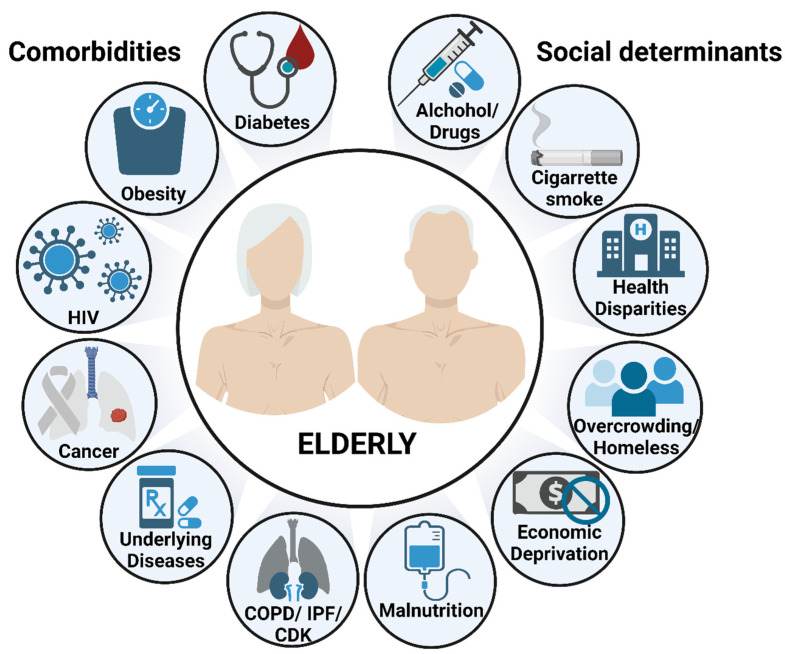
Risk factors of TB disease and mortality in the elderly. Aging by itself is a major risk factor to develop and succumb to TB, partially due to the waning immunity that characterizes the elderly population. Additional risk factors, including comorbidities and socioeconomic determinants, are key drivers to increase even more the susceptibility of the elderly to respiratory infections. This illustration was created with BioRender (https://biorender.com/), accessed on 18 September 2022.

## Data Availability

Not applicable.

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
