# Peer review of "Tuberculosis in an Aging World"

_pathogens, 2022, doi:10.3390/pathogens11101101_

Round 1

Reviewer 1 Report

Changes in populational age structure due to urbanization and other socioeconomic factors is an accelerating and largely irreversible trend that manifests in increasing number of regions in the world, which raises the necessity of rethinking the basic and clinical TB research with the premise that much of what we learnt about TB from adult, immune-competent human or animal models may fail to apply on aged populations. In this draft, Olmo- Fontánez and Turner discuss the impact of aging on tuberculosis infection, disease progression, and treatment as well as other factors of great relevance. The authors have addressed the functional, structural, mechanical, and immunological changes to an aging lung that potentially ally with TB, with a more focused discussion on aging-associated immunological manifestations and their shares in sabotaging normal immune responses to TB infection. The authors have also elaborated on TB comorbidities, including diabetes, malnutrition, and other chronic conditions that are prevalent in the elderly population and further compromise the patients’ immune system and render them unarmed against TB. The manuscript is generally well written and referenced and has addressed some of the important questions/disputes in TB research, which makes it a great fit for the journal Pathogens and is likely found interesting by researchers in the TB field and beyond. Though to be honest I believe the manuscript could be published with just a few language tweaks, I do have a couple comments/suggestions for the authors:

Figure 1 has a wordy and information-rich legend, yet the figure itself doesn’t quite convey the full message. To me, the cellular changes in an aging lung is of great interest and the corresponding directionalities, for instance, the proportional increase of CD11c+ CD11b+ subpopulation or the higher background level of IL1b, would be more visually appealing if they were highlighted by arrows or color warmth. Moreover, this figure failed to address the interplay between these cellular changes and TB control, which is the focus of the corresponding section. 

Line 123-124: here the authors stated that aged inflammatory macrophages are more tolerant to mtb growth and survival and cited reference # 42. The CD11b+ AM subset, according to ref #42, seemed to be phagocytosing more mtb bacilli (2-24hours), but there lacks data in this article to support that these bacteria would exhibit any survival/growth advantage against mtb cells engulfed by CD11b- AMs. Also, this study did not evaluate long term survival of CD11b+ cells after mtb infection, so it’s hard to tell whether this subpopulation can actually “tolerate” mtb or they just have an increased appetite for mtb. I’d suggest the authors to soften the language and call these aged macrophages more “permissive” to mtb infection, as implied in the title of #42.

Line 158-159: the authors stated that the effectiveness of BCG booster in TB protection is dampened in aged mouse, and cited reference #62 (Ozeki et. al., Vaccine, 2011?). I don’t think that the referenced article supports such notion. Ozeki’s work evaluated post-vaccination protection by measuring the clearance of Kan-resistant BCG seeded after primary Kan-sensitive BCG vaccination. Protection against BCG infection is quite far from being efficacious against TB, moreover, their paper, to my knowledge, did not test the efficacy of BCG booster shots, and their data suggested that protection rendered by BCG vaccination is almost equally pronounced in young and old mice, though it vanishes over time. I’m guessing that the authors intended to cite another paper, if so, please rectify. 

Line 284-286: I wanted to point out that reference #107 only tested cytokine/chemokine and other immune profiles w/ or w/o BCG vaccination as a sidetrack of a COVID-19 cohort. This paper did not evaluate protection and in its sibling study and the ACTIVATE trial, the efficacy of BCG (re-)vaccination was only tested against non-TB infectious agents. Please either specify the panel of pathogens that have been tested for the elderly who received BCG (re-)vaccination or cite a different paper.

There’re two recent papers that reported some interesting interplays between TB infection and aging: Bobak and others showed that TB infection might accelerate aging via epigenetic reprogramming (PMID: 35256539); Abbara and others have revealed in a retrospective study that there’s a significant delay in the clinical TB diagnosis for the elderlies (31720296). These studies may provide some alternative perspectives to the general idea of aging being a risk factor for TB (e.g., could TB be a catalyst of aging?), though it’s entirely up to the authors to decide whether this is of high relevance. 

Line 139: “…with advanced aged…” should be “…with advanced aging…”

Reviewer 2 Report

The review is very interesting but I recommend adding a partition focusing on the recent techniques for diagnosis of tuberculosis.

1-The review discussed the risk of TB susceptibility and mortality is significantly increased in individuals aged 65 and older. It focused primarily on why the elderly are more susceptible to TB disease and death with a focus on pulmonary function and comorbidities.

2-The review is relevant and interesting.

3-TB is a notifiable zoonotic disease and causes severe losses, especially after emerging of MDR strains. So the field needs a lot of researches and articles concerning tuberculosis. I suggested that the authors should declare the recent techniques for diagnosis of tuberculosis besides the dangerous effect of MDR strains.

4-Compared to other published material, this review takes a different way in collecting data about strategies that reduce age-related inflammation to improve resistance to TB disease and quality of life in the elderly.

5-The paper is well written.

6-The text is clear and easy to read.

7-The conclusions consistent with the evidence and arguments
presented.

8-The authors addressed the main question posed.
